# Rethinking Numerical Representations for Deep Neural Networks

**Parker Hill, Babak Zamirai, Shengshuo Lu, Yu-Wei Chao, Michael Laurenzano, Mehrzad Samadi**
**Marios Papaefthymiou, Scott Mahlke, Thomas Wenisch, Jia Deng, Lingjia Tang, Jason Mars**
Department of Electrical Engineering and Computer Science
University of Michigan, Ann Arbor
`{parkerhh,zamirai,luss,ywchao,mlaurenz,mahrzads}@umich.edu`
`{marios,mahlke,twenisch,jiadeng,lingjia,profmars}@umich.edu`

## ABSTRACT

With ever-increasing computational demand for deep learning, it is critical to investigate the implications of the numeric representation and precision of DNN model weights and activations on computational efficiency. In this work, we explore unconventional narrow-precision floating-point representations as it relates to inference accuracy and efficiency to steer the improved design of future DNN platforms. We show that inference using these custom numeric representations on production-grade DNNs, including GoogLeNet and VGG, achieves an average speedup of $7.6\times$ with less than 1% degradation in inference accuracy relative to a state-of-the-art baseline platform representing the most sophisticated hardware using single-precision floating point. To facilitate the use of such customized precision, we also present a novel technique that drastically reduces the time required to derive the optimal precision configuration.

## 1 INTRODUCTION

Recently, deep neural networks (DNNs) have yielded state-of-the-art performance on a wide array of AI tasks, including image classification Krizhevsky et al. (2012), speech recognition Hannun et al. (2014), and language understanding Sutskever et al. (2014). In addition to algorithmic innovations Nair & Hinton (2010); Srivastava et al. (2014); Taigman et al. (2014), a key driver behind these successes are advances in computing infrastructure that enable large-scale deep learning—the training and inference of large DNN models on massive datasets Dean et al. (2012); Farabet et al. (2013). Indeed, highly efficient GPU implementations of DNNs played a key role in the first breakthrough of deep learning for image classification Krizhevsky et al. (2012). Given the ever growing amount of data available for indexing, analysis, and training, and the increasing prevalence of ever-larger DNNs as key building blocks for AI applications, it is critical to design computing platforms to support faster, more resource-efficient DNN computation.

A set of core design decisions are common to the design of these infrastructures. One such critical choice is the numerical representation and precision used in the implementation of underlying storage and computation. Several recent works have investigated the numerical representation for DNNs Cavigelli et al. (2015); Chen et al. (2014); Du et al. (2014); Muller & Indiveri (2015). One recent work found that substantially lower precision can be used for training when the correct numerical rounding method is employed Gupta et al. (2015). Their work resulted in the design of a very energy-efficient DNN platform.

This work and other previous numerical representation studies for DNNs have either limited themselves to a small subset of the customized precision design space or drew conclusions using only small neural networks. For example, the work from Gupta et al. 2015 evaluates 16-bit fixed-point and wider computational precision on LeNet-5 LeCun et al. (1998) and CIFARNET Krizhevsky & Hinton (2009). The fixed-point representation (Figure 1) is only one of many possible numeric representations. Exploring a limited customized precision design space inevitably results in designs lacking in energy efficiency and computational performance. Evaluating customized precision accuracy based on small neural networks requires the assumption that much larger, production-grade neural networks would operate comparably when subjected to the same customized precision.

In this work, we explore the accuracy-efficiency trade-off made available via specialized custom-precision hardware for inference and present a method to efficiently traverse this large design space to find an optimal design. Specifically, we evaluate the impact of a wide spectrum of customized

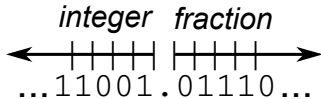

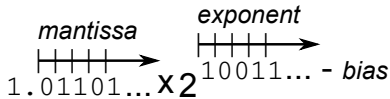

Figure 1: A fixed-point representation. Hardware parameters include the total number of bits and the position of the radix point.

Figure 2: A floating-point representation. Hardware parameters include the number of mantissa and exponent bits, and the bias.

precision settings for fixed-point and floating-point representations on accuracy and computational performance. We evaluate these customized precision configurations on large, state-of-the-art neural networks. By evaluating the full computational precision design space on a spectrum of these production-grade DNNs, we find that:

1. Precision requirements do not generalize across all neural networks. This prompts designers of future DNN infrastructures to carefully consider the applications that will be executed on their platforms, contrary to works that design for large networks and evaluate accuracy on small networks Cavigelli et al. (2015); Chen et al. (2014).

2. Many large-scale DNNs require considerably more precision for fixed-point arithmetic than previously found from small-scale evaluations Cavigelli et al. (2015); Chen et al. (2014); Du et al. (2014). For example, we find that GoogLeNet requires on the order of 40 bits when implemented with fixed-point arithmetic, as opposed to less than 16 bits for LeNet-5.

3. Floating-point representations are more efficient than fixed-point representations when selecting optimal precision settings. For example, a 17-bit floating-point representation is acceptable for GoogLeNet, while over 40 bits are required for the fixed-point representation – a more expensive computation than the standard single precision floating-point format. Current platform designers should reconsider the use of the floating-point representations for DNN computations instead of the commonly used fixed-point representations Cavigelli et al. (2015); Chen et al. (2014); Du et al. (2014); Muller & Indiveri (2015).

To make these conclusions on large-scale customized precision design readily actionable for DNN infrastructure designers, we propose and validate a novel technique to quickly search the large customized precision design space. This technique leverages the activations in the last layer to build a model to predict accuracy based on the insight that these activations effectively capture the propagation of numerical error from computation. Using this method on deployable DNNs, including GoogLeNet Szegedy et al. (2015) and VGG Simonyan & Zisserman (2014), we find that using these recommendations to introduce customized precision into a DNN accelerator fabric results in an average speedup of $7.6\times$ with less than 1% degradation in inference accuracy.

## 2 CUSTOMIZED PRECISION HARDWARE

We begin with an overview of the available design choices in the representation of real numbers in binary and discuss how these choices impact hardware performance.

### 2.1 DESIGN SPACE

We consider three aspects of customized precision number representations. First, we contrast the high-level choice between fixed-point and floating-point representations. Fixed-point binary arithmetic is computationally identical to integer arithmetic, simply changing the interpretation of each bit position. Floating-point arithmetic, however, represents the sign, mantissa, and exponent of a real number separately. Floating-point calculations involve several steps absent in integer arithmetic. In particular, addition operations require aligning the mantissas of each operand. As a result, floating-point computation units are substantially larger, slower, and more complex than integer units.

In CPUs and GPUs, available sizes for both integers and floating-point calculations are fixed according to the data types supported by the hardware. Thus, the second aspect of precision customization we examine is to consider customizing the number of bits used in representing floating-point and fixed-point numbers. Third, we may vary the interpretation of fixed-point numbers and assignment of bits to the mantissa and exponent in a floating-point value.

### 2.2 CUSTOMIZED PRECISION TYPES

In a fixed-point representation, we select the number of bits as well as the position of the radix point, which separates integer and fractional bits, as illustrated in Figure 1. A bit array, $x$, encoded in fixed point with the radix point at bit $l$ (counting from the right) represents the value $2^{-l} \sum_{i=0}^{N-1} 2^i \cdot x_i$.

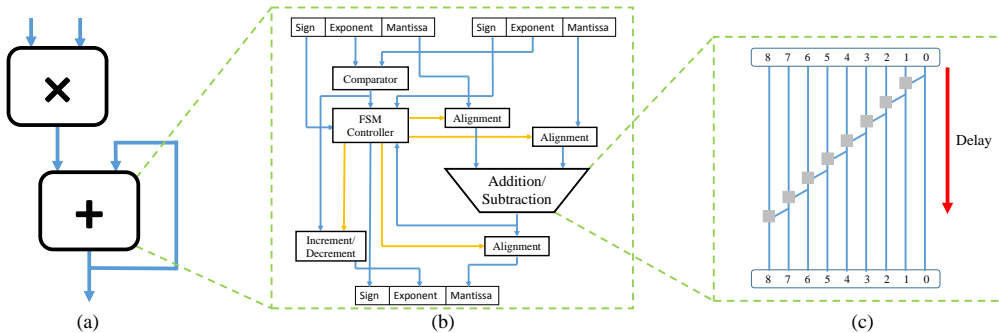

Figure 3: Floating point multiply-accumulate (MAC) unit with various levels of detail: (a) the high level mathematical operation, (b) the modules that form a floating point MAC, and (c) the signal propagation of the unit.

In contrast to floating point, fixed-point representations with a particular number of bits have a fixed level of precision. By varying the position of the radix point, we change the representable range.

An example floating-point representation is depicted in Figure 2. As shown in the figure, there are three parameters to select when designing a floating-point representation: the bit-width of the mantissa, the bit-width of the exponent, and an exponent bias. The widths of the mantissa and exponent control precision and dynamic range, respectively. The exponent bias adjusts the offset of the exponent (which is itself represented as an unsigned integer) relative to zero to facilitate positive and negative exponents. Finally, an additional bit represents the sign. Thus, a floating-point format with $N_m$ mantissa bits, $N_e$ exponent bits, and a bias of $b$, encodes the value $2^{(\sum_{i=0}^{N_e-1} 2^i \cdot e_i)-b}(1 + \sum_{i=1}^{N_m} 2^{-i} \cdot m_i)$, where $m$ and $e$ are the segments of a bit array representing the mantissa and exponent, respectively. Note that the leading bit of the mantissa is assumed to be 1 and hence is not explicitly stored, eliminating redundant encodings of the same value. A single-precision value in the IEEE-754 standard (i.e. `float`) comprises 23 mantissa bits, 8 exponent bits, and a sign bit. IEEE-754 standardized floating-point formats include special encodings for specific values, such as zero and infinity.

Both fixed-point and floating-point representations have limitations in terms of the precision and the dynamic ranges available given particular representations, manifesting themselves computationally as rounding and saturation errors. These errors propagate through the deep neural network in a way that is difficult to estimate holistically, prompting experimentation on the DNN itself.

## 2.3 Hardware Implications

The key hardware building block for implementing DNNs is the multiply-accumulate (MAC) operation. The MAC operation implements the sum-of-products operation that is fundamental to the activation of each neuron. We show a high-level hardware block diagram of a MAC unit in Figure 3 (a). Figure 3 (b) adds detail for the addition operation, the more complex of the two operations. As seen in the figure, floating-point addition operations involve a number of sub-components that compare exponents, align mantissas, perform the addition, and normalize the result. Nearly all of the sub-components of the MAC unit scale in speed, power, and area with the bit width.

Reducing the floating-point bit width improves hardware performance in two ways. First, reduced bit width makes a computation unit faster. Binary arithmetic computations involve chains of logic operations that typically grows at least logarithmically, and sometimes linearly (e.g., the propagation of carries in an addition, see Figure 3 (c)), in the number of bits. Reducing the bit width reduces the length of these chains, allowing the logic to operate at a higher clock frequency. Second, reduced bit width makes a computation unit smaller and require less energy, typically linearly in the number of bits. The circuit delay and area is shown in Figure 4 when the mantissa bit widths are varied. As shown in the figure, scaling the length of the mantissa provides substantial opportunity because it defines the size of the internal addition unit. Similar trends follow for bit-widths in other representations. When a unit is smaller, more replicas can fit within the same chip area and power budget, all of which can operate in parallel. Hence, for computations like those in DNNs, where ample parallelism is available, area reductions translate into proportional performance improvement.

This trend of bit width versus speed, power, and area is applicable to every computation unit in hardware DNN implementations. Thus, in designing hardware that uses customized representations

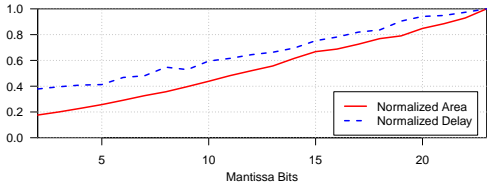
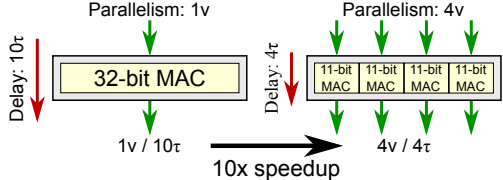

Figure 4: Delay and area implications of mantissa width, normalized to a 32-bit Single Precision MAC with 23 mantissa bits.

Figure 5: Speedup calculation with a fixed area budget. The speedup exploits the improved function delay and parallelism.

there is a trade-off between accuracy on the one hand and power, area, and speed on the other. Our goal is to use precision that delivers sufficient accuracy while attaining large improvements in power, area, and speed over standard floating-point designs.

## 3 METHODOLOGY

We describe the methodology we use to evaluate the customized precision design space, using image classification tasks of varying complexity as a proxy for computer vision applications. We evaluate DNN implementations using several metrics, classification accuracy, speedup, and energy savings relative to a baseline custom hardware design that uses single-precision floating-point representations. Using the results of this analysis, we propose and validate a search technique to efficiently determine the correct customized precision design point.

### 3.1 ACCURACY

We evaluate accuracy by modifying the Caffe Jia et al. (2014) deep learning framework to perform calculations with arbitrary fixed-point and floating-point formats. We continue to store values as C `floats` in Caffe, but truncate the mantissa and exponent to the desired format after each arithmetic operation. Accuracy, using a set of test inputs disjoint from the training input set, is then measured by running the forward pass of a DNN model with the customized format and comparing the outputs with the ground truth. We use the standard accuracy metrics that accompany the dataset for each DNN. For MNIST (LeNet-5) and CIFAR-10 (CIFARNET) we use top-1 accuracy and for ImageNet (GoogLeNet, VGG, and AlexNet) we use top-5 accuracy. Top-1 accuracy denotes the percent of inputs that the DNN predicts correctly after a single prediction attempt, while top-5 accuracy represents the percent of inputs that DNN predicts correctly after five attempts.

### 3.2 EFFICIENCY

We quantify the efficiency advantages of customized floating-point representations by designing a floating-point MAC unit in each candidate precision and determining its silicon area and delay characteristics. We then report speedup and energy savings relative to a baseline custom hardware implementation of a DNN that uses standard single-precision floating-point computations. We design each variant of the MAC unit using Synopsys Design Compiler and Synopsys PrimeTime, industry standard ASIC design tools, targeting a commercial 28nm silicon manufacturing process. The tools report the power, delay, and area characteristics of each precision variant. As shown in Figure 5, we compute speedups and energy savings relative to the standardized IEEE-754 floating-point representation considering both the clock frequency advantage and improved parallelism due to area reduction of the narrower bit-width MAC units. This allows customized precision designs to yield a quadratic improvement in total system throughput.

### 3.3 EFFICIENT CUSTOMIZED PRECISION SEARCH

To exploit the benefits of customized precision, a mechanism to select the correct configuration must be introduced. There are hundreds of designs among floating-point and fixed-point formats due to designs varying by the total bit width and the allocation of those bits. This spectrum of designs strains the ability to select an optimal configuration. A straightforward approach to select the customized precision design point is to exhaustively compute the accuracy of each design with a large number of neural network inputs. This strategy requires substantial computational resources that are proportional to the size of the network and variety of output classifications. We describe our technique that significantly reduces the time required to search for the correct configuration in order to facilitate the use of customized precision.

The key insight behind our search method is that customized precision impacts the underlying internal computation, which is hidden by evaluating only the NN final accuracy metric. Thus, instead

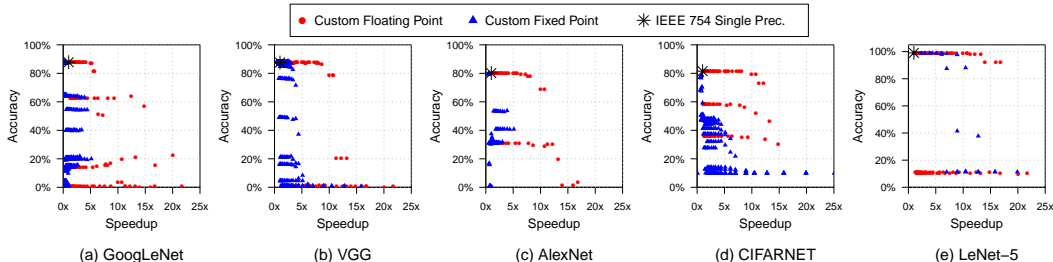

Figure 6: The inference accuracy versus speedup design space for each of the neural networks, showing substantial computational performance improvements for minimal accuracy degradation when customized precision floating-point formats are used.

of comparing the final accuracy generated by networks with different precision configurations, we compare the original NN activations to the customized precision activations. This circumvents the need to evaluate the large number of inputs required to produce representative neural network accuracy. Furthermore, instead of examining all of the activations, we only analyze the last layer, since the last layer captures the usable output from the neural network as well as the propagation of lost accuracy. Our method summarizes the differences between the last layer of two configurations by calculating the linear coefficient of determination between the last layer activations.

A method to translate the coefficient of determination to a more desirable metric, such as end-to-end inference accuracy, is necessary. We find that a linear model provides such a transformation. The customized precision setting with the highest speedup that meets a specified accuracy threshold is then selected. In order to account for slight inaccuracies in the model, inference accuracy for a subset of configurations is evaluated. If the configuration provided by the accuracy model results in insufficient accuracy, then an additional bit is added and the process repeats. Similarly, if the accuracy threshold is met, then a bit is removed from the customized precision format.

## 4 EXPERIMENTS

In this section, we evaluate five common neural networks spanning a range of sizes and depths in the context of customized precision hardware. We explore the trade-off between accuracy and efficiency when various customized precision representations are employed. Next, we address the sources of accuracy degradation when customized precision is utilized. Finally, we examine the characteristics of our customized precision search technique.

### 4.1 EXPERIMENTAL SETUP

We evaluate the accuracy of customized precision operations on five DNNs: GoogLeNet Szegedy et al. (2015), VGG Simonyan & Zisserman (2014), AlexNet Krizhevsky et al. (2012), CIFAR-NET Krizhevsky & Hinton (2009), and LeNet-5 LeCun et al. (1998). The implementations and pre-trained weights for these DNNs were taken from Caffe Jia et al. (2014). The three largest DNNs (GoogLeNet, VGG, and AlexNet) represent real-world workloads, while the two smaller DNNs (CIFARNET and LeNet-5) are the largest DNNs evaluated in prior work on customized precision. For each DNN, we use the canonical benchmark validation set: ImageNet for GoogLeNet, VGG, and AlexNet; CIFAR-10 for CIFARNET; MNIST for LeNet-5. We utilize the entire validation set for all experiments, except for GoogLeNet and VGG experiments involving the entire design space. In these cases we use a randomly-selected 1% of the validation set to make the experiments tractable.

### 4.2 ACCURACY VERSUS EFFICIENCY TRADE-OFFS

To evaluate the benefits of customized precision hardware, we swept the design space for accuracy and performance characteristics. This performance-accuracy trade off is shown in Figure 6. This figure shows the DNN inference accuracy across the full input set versus the speedup for each of the five DNN benchmarks. The black star represents the IEEE 754 single precision representation (i.e. the original accuracy with $1\times$ speedup), while the red circles and blue triangles represent the complete set of our customized precision floating-point and fixed-point representations, respectively.

For GoogLeNet, VGG, and AlexNet it is clear that the floating-point format is superior to the fixed-point format. In fact, the standard single precision floating-point format is faster than all fixed-point configurations that achieve above 40% accuracy. Although fixed-point computation is simpler and faster than floating-point computation when the number of bits is fixed, customized precision floating-point representations are more efficient because less bits are needed for similar accuracy.

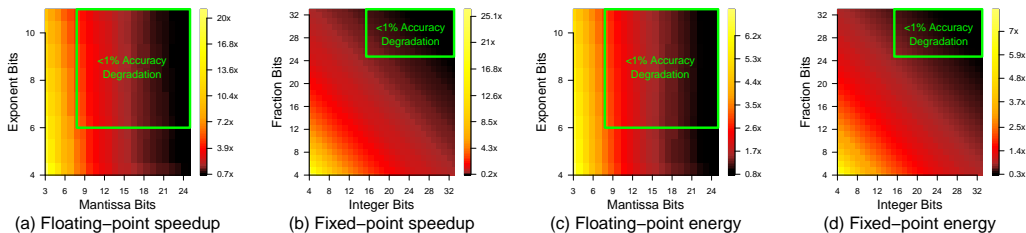

Figure 7: The speedup and energy savings as the two parameters are adjusted for the custom floating point and fixed-point representations. The marked area denotes configurations where the total loss in AlexNet accuracy is less than 1%.

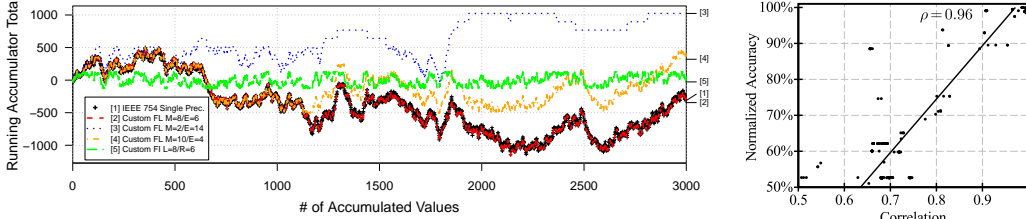

Figure 8: The accumulation of weighted neuron inputs for a specific neuron with various customized precision DNNs as well as the IEEE 754 single precision floating point configuration for reference. FL and FI are used to abbreviate floating point and fixed-point, respectively. The format parameters are as follows: M=mantissa, E=exponent, L=bits left of radix point, R=bits right of radix point.

Figure 9: The linear fit from the correlation between normalized accuracy and last layer activations of the exact and customized precision DNNs.

By comparing the results across the five different networks in Figure 6, it is apparent that the size and structure of the network impacts the customized precision flexibility of the network. This insight suggests that hardware designers should carefully consider which neural network(s) they expect their device to execute as one of the fundamental steps in the design process. The impact of network size on accuracy is discussed in further detail in the following section.

The specific impact of bit assignments on performance and energy efficiency are illustrated in Figure 7. This figure shows the the speedup and energy improvements over the single precision floating-point representation as the number of allocated bits is varied. For the floating-point representations, the number of bits allocated for the mantissa (x-axis) and exponent (y-axis) are varied. For the fixed-point representations, the number of bits allocated for the integer (x-axis) and fraction (y-axis) are varied. We highlight a region in the plot deemed to have acceptable accuracy. In this case, we define acceptable accuracy to be 99% normalized AlexNet accuracy (i.e., no less than a 1% degradation in accuracy from the IEEE 754 single precision accuracy on classification in AlexNet).

The fastest and most energy efficient representation occurs at the bottom-left corner of the region with acceptable accuracy, since a minimal number of bits are used. The configuration with the highest performance that meets this requirement is a floating-point representation with 6 exponent bits and 7 mantissa bits, which yields a $7.2\times$ speedup and a $3.4\times$ savings in energy over the single precision IEEE 754 floating-point format. If a more stringent accuracy requirement is necessary, 0.3% accuracy degradation, the representation with one additional bit in the mantissa can be used, which achieves a $5.7\times$ speedup and $3.0\times$ energy savings.

### 4.3 SOURCES OF ACCUMULATION ERROR

In order to understand how customized precision degrades DNN accuracy among numeric representations, we examine the impact of various reduced precision computations on a neuron. Figure 8 presents the serialized accumulation of neuron inputs in the third convolution layer of AlexNet. The x-axis represents the number of inputs that have been accumulated, while the y-axis represents the current value of the running sum. The black line represents the original DNN computation, a baseline for customized precision settings to match. We find two causes of error between the customized precision fixed-point and floating-point representations, saturation and excessive rounding.

In the fixed-point case (green line, representing 16 bits with the radix point in the center), the central cause of error is from saturation at the extreme values. The running sum exceeds 255, the maximum representable value in this representation, after 60 inputs are accumulated, as seen in the figure.

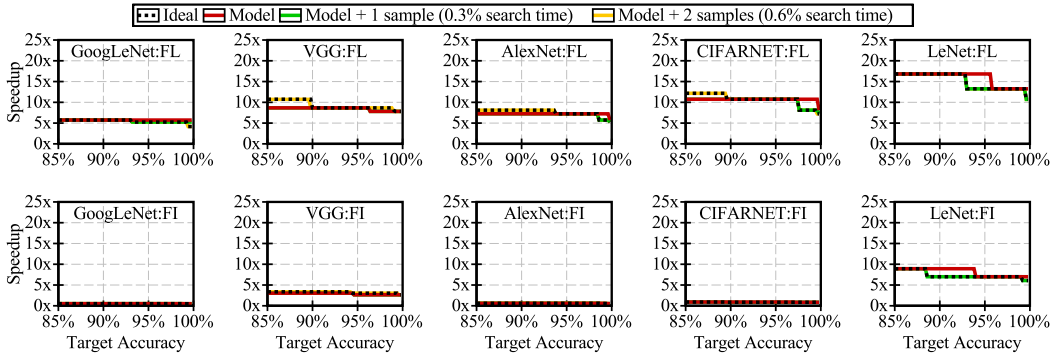

Figure 10: The speedup achieved by selecting the customized precision using an exhaustive search (i.e. the ideal design) and prediction using the accuracy model with accuracy evaluated for some number of configurations (model + X samples). The floating-point (FL) and fixed-point (FI) results are shown in the top and bottom rows, respectively. The model with two evaluated designs produces the same configurations, but requires <0.6% of the search time.

After reaching saturation, the positive values are discarded and the final output is unpredictable. Although floating-point representations do not saturate as easily, the floating-point configuration with 10 mantissa bits and 4 exponent bits (orange line) saturates after accumulating 1128 inputs. Again, the lost information from saturation causes an unpredictable final output.

For the next case, the floating-point configuration with 2 bits and 14 bits for the mantissa and exponent (blue line), respectively, we find that the lack of precision for large values causes excessive rounding errors. As shown in the figure, after accumulating 120 inputs, this configuration's running sum exceeds 256, which limits the minimum adjustment in magnitude to 64 (the exponent normalizes the mantissa to 256, so the two mantissa bits represent 128 and 64). Finally, one of the customized precision types that has high performance and accuracy for AlexNet, 8 mantissa bits and 6 exponent bits (red line), is shown as well. This configuration almost perfectly matches the IEEE 754 floating-point configuration, as expected based on the final output accuracy.

The other main cause of accuracy loss is from values that are too small to be encoded as a non-zero value in the chosen customized precision configuration. These values, although not critical during addition, cause significant problems when multiplied with a large value, since the output should be encoded as a non-zero value in the specific precision setting. We found that the weighted input is minimally impacted, until the precision is reduced low enough for the weight to become zero.

While it may be intuitive based on these results to apply different customized precision settings to various stages of the neural network in order to mitigate the sudden loss in accuracy, the realizable gains of multi-precision configurations present significant challenges. The variability between units will cause certain units to be unused during specific layers of the neural network causing gains to diminish (e.g., 11-bit units are idle when 16-bit units are required for a particular layer). Also, the application specific hardware design is already an extensive process and multiple customized precision configurations increases the difficulty of the hardware design and verification process.

## 4.4 CUSTOMIZED PRECISION SEARCH

Now we evaluate our proposed customized precision search method. The goal of this method is to significantly reduce the required time to navigate the customized precision design space and still provide an optimal design choice in terms of speedup, limited by an accuracy constraint.

**Correlation model.** First, we present the linear correlation-accuracy model in Figure 9, which shows the relationship between the normalized accuracy of each setting in the design space and the correlation between its last layer activations compared to those of the original NN. This model, although built using all of the customized precision configurations from AlexNet, CIFARNET, and LeNet-5 neural networks, produces a good fit with a correlation of 0.96. It is important that the model matches across networks and precision design choices (e.g., floating point versus fixed point), since creating this model for each DNN, individually, requires as much time as exhaustive search.

**Validation.** To validate our search technique, Figure 10 presents the accuracy-speedup trade-off curves from our method compared to the ideal design points. We first obtain optimal results via

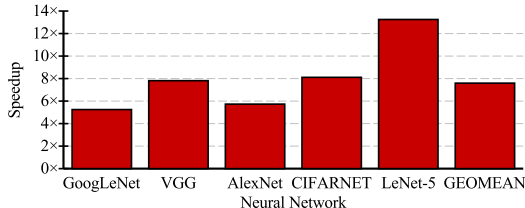

Figure 11: The speedup resulting from searching for the fastest setting with less than 1% inference accuracy degradation. All selected customized precision DNNs meet this accuracy constraint.

exhaustive search. We present our search with a variable number of refinement iterations, where we evaluate the accuracy of the current design point and adjust the precision if necessary. To verify robustness, the accuracy models were generated using cross-validation where all configurations in the DNN being searched are excluded (e.g., we build the AlexNet model with LeNet and CIFAR-NET accuracy/correlation pairs). The prediction is made using only ten randomly selected inputs, a tiny subset compared that needed for classification accuracy, some of which are even incorrectly classified by the original neural network. Thus, the cost of prediction using the model is negligible.

We observe that, in all cases, the accuracy model combined with the evaluation of just two customized precision configurations provides the same result as the exhaustive search. Evaluating two designs out of 340 is 170× faster than exhaustively evaluating all designs. When only one configuration is evaluated instead of two (i.e. a further 50% reduction is search time), the selected customized precision setting never violates the target accuracy, but concedes a small amount of performance. Finally, we note that our search mechanism, without evaluating inference accuracy for any of the design points, provides a representative prediction of the optimal customized precision setting. Although occasionally violating the target accuracy (i.e. the cases where the speedup is higher than the exhaustive search), this prediction can be used to gauge the amenability of the NN to customized precision without investing any considerable amount of time in experimentation.

**Speedup.** We present the final speedup produced by our search method in Figure 11 when the algorithm is configured for 99% target accuracy and to use two samples for refinement. In all cases, the chosen customized precision configuration meets the targeted accuracy constraint. In most cases, we find that the larger networks require more precision (DNNs are sorted from left to right in descending order based on size). VGG requires less precision than expected, but VGG also uses smaller convolution kernels than all of the other DNNs except LeNet-5.

## 5 RELATED WORK

To the best of our knowledge, our work is the first to examine the impact of numeric representations on the accuracy-efficiency trade-offs on large-scale, deployed DNNs with over half a million neurons (GoogLeNet, VGG, AlexNet), whereas prior work has only reported results on much smaller networks such as CIFARNET and LeNet-5 Cavigelli et al. (2015); Chen et al. (2014); Courbariaux et al. (2014); Du et al. (2014); Gupta et al. (2015); Muller & Indiveri (2015). Many of these works focused on fixed-point computation due to the fixed-point representation working well on small-scale neural networks. We find very different conclusions when considering production-ready DNNs.

Other recent works have looked at alternative neural network implementations such as spiking neural networks for more efficient hardware implementation Conti & Benini (2015); Diehl & Cook (2014). This is a very different computational model that requires redevelopment of standard DNNs, unlike our proposed methodologies. Other works have proposed several approaches to improve performance and reduce energy consumption of deep neural networks by taking advantage of the fact that DNNs usually contain redundancies Chen et al. (2015); Figurnov et al. (2015).

## 6 CONCLUSION

In this work, we introduced the importance of carefully considering customized precision when realizing neural networks. We show that using the IEEE 754 single precision floating point representation in hardware results in surrendering substantial performance. On the other hand, picking a configuration that has lower precision than optimal will result in severe accuracy loss. By reconsidering the representation from the ground up in designing custom precision hardware and using our search technique, we find an average speedup across deployable DNNs, including GoogLeNet and VGG, of 7.6× with less than 1% degradation in inference accuracy.

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
