# Peer review of "Rethinking Numerical Representations for Deep Neural Networks"

_ICLR 2017 — rejected_

[Official Review · AnonReviewer2 · rating 5 · confidence 2 · 19 Dec 2016]
**Can be improved**

The paper provides a first study of customized precision hardware for large convolutional networks, namely alexnet, vgg and googlenet. It shows that it is possible to achieve larger speed-ups using floating-point precision (up to 7x) when using fewer bits, and better than using fixed-point representations. 

The paper also explores predicting custom floating-point precision parameters directly from the neural network activations, avoiding exhaustive search, but i could not follow this part. Only the activations of the last layer are evaluated, but on what data ? On all the validation set ? Why would this be faster than computing the classification accuracy ?

The results should be useful for hardware manufacturers, but with a catch. All popular convolutional networks now use batch normalization, while none of the evaluated ones do. It may well be that the conclusions of this study will be completely different on batch normalization networks, and fixed-point representations are best there, but that remains to be seen. It seems like something worth exploring.

Overall there is not a great deal of novelty other than being a useful study on numerical precision trade-offs at neural network test time. Training time is also something of interest. There are a lot more researchers trying to train new networks fast than trying to evaluate old ones fast. 

I am also no expert in digital logic design, but my educated guess is that this paper is marginally below the acceptance threshold.

[Official Review · AnonReviewer5 · rating 5 · confidence 5 · 27 Dec 2016 (modified: 02 Jan 2017)]
**Ignores broader system-level issues, needs to use 16-bit floats as baseline**

This paper explores the performance-area-energy-model accuracy tradeoff encountered in designing custom number representations for deep learning inference. Common image-based benchmarks: VGG, Googlenet etc are used to demonstrate that fewer than1 6 bits in a custom floating point representation can lead to improvement in runtime performance and energy efficiency with only a small loss in model accuracy.

Questions:

1. Does the custom floating point number representation take into account support for de-normal numbers? 
2. Is the custom floating point unit clocked at the same frequency as the baseline 32-bit floating point unit? If not, what are the different frequencies used and how would this impact the overall system design in terms of feeding the data to the floating point units from the memory

Comments:

1. I would recommend using the IEEE half-precision floating point (1bit sign, 5bit exponent, and 10bit mantissa) as a baseline for comparison. At this point, it is well known in both the ML and the HW communities that 32-bit floats are an overkill for DNN inference and major HW vendors already include support for IEEE half-precision floats. 
2. In my opinion, the claim that switching to custom floating point  lead to a YY.ZZ x savings in energy is misleading. It might be true that the floating-point unit itself might consume less energy due to smaller bit-width of the operands, however a large fraction of the total energy is spent in data movement to/from the memories. As a result, reducing the floating point unit’s energy consumption by a certain factor will not translate to the same reduction in the total energy. A reader not familiar with such nuances (for example a typical member of the ML community), may be mislead by such claims. 
3. On a similar note as comment 2, the authors should explicitly mention that the claimed speedup is that of the floating point unit only, and it will not translate to the overall workload speedup. Although the speedup of the compute unit is roughly quadratic in the bit-width, the bandwidth requirements scale linearly with bit-width. As a result, it is possible that these custom floating point units may be starved on memory bandwidth, in which case the claims of speedup and energy savings need to be revisited.
4. The authors should also comment on the complexities and overheads introduced in data accesses, designing the various system buses/ data paths when the number representation is not byte-aligned. Moving to a custom 14-bit number representation (for example) can improve the performance and energy-efficiency of the floating point unit, but these gains can be partially eroded due to the additional overhead in supporting non-byte aligned memory accesses.

[Official Review · AnonReviewer3 · rating 6 · confidence 3 · 03 Jan 2017]

The paper studies the impact of using customized number representations on accuracy, speed, and energy consumption of neural network inference. Several standard computer vision architectures including VGG and GoogleNet are considered for the experiments, and it is concluded that floating point representations are preferred over fixed point representations, and floating point numbers with about 14 bits are sufficient for the considered architectures resulting in a small loss in accuracy.

The paper provides a nice overview of floating and fixed point representations and focuses on an important aspect of deep learning that is not well studied. There are several aspects of the paper that could be improved, but overall, I am leaned toward weak accept assuming that the authors address the issues below.

1- The paper is not clear that it is only focusing on neural network inference. Please include the word "inference" in the title / abstract to clarify this point and mention that the findings of the paper do not necessarily apply to neural network training as training dynamics could be different.

2- The paper does not discuss the possibility of adopting quantization tricks during training, which may result in the use of fewer bits at inference.

3- The paper is not clear whether in computing the running time and power consumption, it includes all of the modules or only multiply-accumulate units? Also, how accurate are these numbers given different possible designs and the potential difference between simulation and production? Please elaborate on the details of simulation in the paper.

4- The whole discussion about "efficient customized precision search" seem unimportant to me. When such important hardware considerations are concerned, even spending 20x simulation time is not that important. The exhaustive search process could be easily parallelized and one may rather spend more time at simulation at the cost of finding the exact best configuration rather than an approximation. That said, weak configurations could be easily filtered after evaluating just a few examples.

5- Nvidia's Pascal GP100 GPU supports FP16. This should be discussed in the paper and relevant Nvidia papers / documents should be cited.

More comments:

- Parts of the paper discussing "efficient customized precision search" are not clear to me.

- As future work, the impact of number representations on batch normalization and recurrent neural networks could be studied.

[Final Decision · Program Chairs · 06 Feb 2017]
**ICLR committee final decision**

The reviewers feel that this is a well written paper on floating and fixed point representations for inference with several state of the art deep learning architectures. At the same time, in order for results to be more convincing, they recommend using 16-bit floats as a more proper baseline for comparison, and to analyze tradeoffs in overall workload speedup, i.e broader system-level issues surrounding the implementation of custom floating point units.